# Magnetron Sputtering Thin Films as Tool to Detect Triclosan in Infant Formula Powder: Electronic Tongue Approach

Cátia Magro [1,2,*], Margarida Sardinha [1], Paulo A. Ribeiro [3], Maria Raposo [3] and Susana Sério [3,*]

1   Department of Physics, NOVA School of Science and Technology, NOVA University Lisbon, 2829-516 Caparica, Portugal; mi.sardinha@campus.fct.unl.pt
2   CENSE—Center for Environmental and Sustainability Research, Department of Environmental Sciences and Engineering, NOVA School of Science and Technology, NOVA University Lisbon, 2829-516 Caparica, Portugal
3   CEFITEC, Department of Physics, NOVA School of Science and Technology, NOVA University Lisbon, 2829-516 Caparica, Portugal; pfr@fct.unl.pt (P.A.R.); mfr@fct.unl.pt (M.R.)
*   Correspondence: c.magro@fct.unl.pt (C.M.); susana.serio@fct.unl.pt (S.S.)

**Abstract:** Triclosan (TCS) is being detected in breast milk and in infants of puerperal women. The harmful effects caused by this compound on living beings are now critical and thus it is pivotal find new tools to TCS monitoring. In the present study, an electronic tongue (e-tongue) device comprising an array of sputtered thin films based on Multi-Walled Carbon Nanotubes and titanium dioxide was developed to identify TCS concentrations, from $10^{-15}$ to $10^{-5}$ M, in both water and milk-based solutions. Impedance spectroscopy was used for device signal transducing and data was analyzed by principal component analysis (PCA). The e-tongue revealed to be able to distinguish water from milk-based matrices through the two Principal Components (PC1 and PC2), which represented 67.3% of the total variance. The PC1 values of infant formula milk powder prepared with tap water (MT) or mineral water (MMW) follows a similar exponential decay curve when plotted with the logarithm of concentration. Therefore, considering the TCS concentration range between $10^{-15}$ and $10^{-9}$ M, the PC1 values are fitted by a straight line and values of $-1.9 \pm 0.2$ and of $7.6 \times 10^{-16}$ M were calculated for the sensor sensitivity and sensor resolution, respectively. Additionally, a strong correlation (R = 0.96) between MT and MMW PC1 data was found. These results have shown that the proposed device corresponds to a promisor method for the detection of TCS in milk-based solutions.

**Keywords:** triclosan; infant formula powder; electronic tongue; thin films; coating; magnetron sputtering; impedance spectroscopy



## 1. Introduction

Triclosan (2,4,4′-Trichloro-2′-hydroxydiphenyl ether) corresponds to an antimicrobial agent that has been utilized for more than five decades as an antiseptic, disinfectant or preservative in clinical situations and numerous consumer products. TCS has been identified in wastewater treatment plants [1–3] and in surface water [4,5]. Recent publications of TCS describe various health effects, including endocrine-disruption, uncoupling mitochondria, among others [6,7]. Additionally, TCS due to its high lipophilicity characteristics, has been detected in breast milk [8], and in babies of those mothers who were exposed to this compound [8,9]. Finally, TCS may also be associated with the progression and growth of breast cancer, due to its estrogenic activity in human cells of this type of cancer [10–12]. Notwithstanding, in 2016, TCS has been banned by US Food and Drug Administration from certain washing products, specifically hand soap and body wash [13] and from hospital products in the end of 2018 [14], however, it is still permitted to include TCS in e.g., toothpaste, cosmetics, clothes or toys [15]. Thus, it becomes relevant to develop and optimize a method for detecting low concentrations of TCS in aqueous solutions or emulsions, in particular, milk-based samples.

Electronic tongues (e-tongues), "a multi-sensory system, formed by an array of low-selective sensor units, combined with advanced mathematical procedures for signal processing based on pattern recognition and/or multi-variate data analysis" [16], have been gaining greater attention in liquid matrices since 1997. There has been an increase in the number of scientific articles, with several applications, although Magro et al., (2019; 2020) [17,18] showed that there is a high potential of e-tongues for qualitative and semi-quantitative multi-analyte analysis in complex matrices using as electrical transducer the impedance spectroscopy. The sensors are composed by two basic components coupled in series: molecular recognition system (receptor) and a physico-chemical transducer [19]. Riul et al., (2002) and Olivati et al., (2009) [20,21] described the use of impedance spectroscopy applied to e-tongues, which showed to be advantageous since the materials comprising the sensing units do not need to be electroactive and a reference electrode is not mandatory, in contrast to the conventional electrochemical methods. In this measurement procedure, the impedance of the entire system is registered for variable frequencies of the signal applied on interdigitated electrodes (IDE) [22]. Furthermore, in the e-tongue's array, the working electrodes may be coated with films, to improve the sensitivity. The capability to adjust the composition of nanostructured thin films allows an enhancement in the sensor's intrinsic chemical or physical properties for sensing applications. Sputtering deposition is a widely used technique to deposit thin films on solid supports. This technique exhibits various advantages such as high deposition rates, high films' purity, high adhesion to the substrate, precise film thickness control and a broad industrial applicability [23–25].

On this base, Multi-Walled Carbon Nanotubes (MWCNTs) and titanium dioxide ($TiO_2$) thin films were studied. $TiO_2$ thin films have been used in electrocatalytic, photocatalytic and photo electrocatalytic degradation of organic pollutants including TCS [26–29]. $TiO_2$ presents adsorption ability to the lipopolysaccharide, due to the high relative surface area, hydrophilic property and electrostatic attraction [30]. Similarly, carbon nanotubes have been employed as catalysts, sensors or displays since these films present great structural, mechanical and electronic properties. MWCNTs are good adsorbent materials with high surface area due to the availability of many active sites at their tip and surface [31]. Hence, the aim of this study is to ascertain the potential of an e-tongue, based in a set of four sensor devices built up with magnetron sputtered thin films, to distinguish different TCS concentrations. Its potentiality is discussed through the principal component analysis of the impedance measurements for different aqueous matrices, with the target matrices being infant formula powder prepared with tap water and mineral water.

## 2. Materials and Methods

In this work, all chemicals employed were in analytical grade or chemical grade (Sigma–Aldrich, Steinheim, Germany).

### 2.1. Materials

The experimental aqueous matrices used to prepare the infant formula were tap water (T) and a Portuguese mineral water (MW). Tap water was collected at Monte de Caparica, Almada, Portugal, and MW was a commercial Portuguese mineral water [17]. The used infant formula powder was Aptamil 1—Milk (Nutricia, Lisbon, Portugal), for infants from 0 to 6 months. The milk-based solutions were prepared according to the water/infant formula powder proportions given by the manufacturer. For convenience of the reader, the infant formula prepared with tap water and with mineral water will be designated by MT and MMW, respectively.

The experimental triclosan (TCS $\geq$ 97%) dilution's range, $10^{-15}$–$10^{-5}$ M, was made in a sequential order from a mother solution with a concentration of $10^{-4}$ M and immediately analyzed after preparation. All dilutions were made using experimental matrices/MeOH (9:1) solutions (methanol-MeOH). A solution of each experimental matrix/MeOH, without TCS, was employed as the blank standard (0 M). Table 1 presents the characterization of the

matrices' characteristics for pH and conductivity parameters, which were accessed with a pH Prolab 1000 BNC Set meter (SI Analytics, Mainz, Germany).

**Table 1.** Matrices characteristics.

| Matrix | pH | Conductivity (mS cm$^{-1}$) | Code |
|---|---|---|---|
| Tap water | $7.06 \pm 0.01$ | $0.48 \pm 0.05$ | T |
| Mineral water | $5.97 \pm 0.01$ | $0.047 \pm 0.005$ | MW |
| Infant formula powder prepared with tap water | $6.89 \pm 0.01$ | $2.0 \pm 0.5$ | MT |
| Infant formula powder prepared with mineral water | $7.11 \pm 0.01$ | $1.54 \pm 0.05$ | MMW |

The sensor devices were acquired from DropSens (Llanera, Asturias, Spain) and they were formed by BK7 glass solid support with deposited interdigitated electrodes (IDE) including 250 or 125 "fingers" with deposited gold IDE comprising eight "fingers" each. The substrates' dimensions were $22.8 \times 7.6 \times 0.7$ mm and each "finger" had 5 or 10 μm of width, which was the same spacing between them.

*2.2. Deposition of MWCNTs-Based Thin Films by RF Magnetron Sputtering*

MWCNTs-based thin films were grown by RF magnetron sputtering using MWCNTs powder (Timesnano, Chengdu Organic Chemicals Co. Ltd., Chinese Academy of Sciences, Chengdu, P.R. China, 95% purity and nanotubes length ~50 μm) as sputtering target. The sputtering target (25 mm diameter and 1 mm of thickness) was prepared by mixing the polycrystalline MWCNTs with acetone. The mixture was placed and compacted until acetone evaporation on the top of a cylindrical body planar magnetron cathode, a prototype that was developed at Plasmas and Applications laboratory to support small powder targets, previously reported and described in detail elsewhere [23]. The sputtering of the target was performed in argon (99.99%, Air Liquide, Paris, France) for different sputtering conditions summarized in Table 2. It was used as a RF power supply: Plasmaloc 2HF (PTB Sales, Azusa, CA, USA) and a frequency of 100 kHz. A turbomolecular pump (CFV 900 turbo Alcatel, Paris, France) was utilized to attain a base pressure of $10^{-4}$ Pa (before introducing the sputtering gas). The target-to-support distance was fixed at 20 mm.

**Table 2.** Sputtering conditions for the deposition of MWCNTs and TiO$_2$ thin films onto BK7 solid support with gold interdigitate electrodes, G-IDEAU5 and G-IDEAU10.

| | Power (W) | Pressure (mbar) | Time (min) | Frequency (kHz) | Impedance (Ω) | | Code |
|---|---|---|---|---|---|---|---|
| G-IDEAU5 | 25 | $2.0 \times 10^{-2}$ | 20 | 100 | 600 | | MWCNT5 |
| G-IDEAU10 | 30 | $2.1 \times 10^{-2}$ | 20 | 100 | 600 | | MWCNT10 |
| | 85 | $2.5 \times 10^{-2}$ | 30 | 100 | 600 | | MWCNT10-85 |
| | | | | Voltage (V) | Current (A) | O$_2$ (%) | |
| G-IDEAU5 | 520 | $2.0 \times 10^{-2}$ | 25 | 458 | 1.14 | 100 | TiO$_2$ |

*2.3. Deposition of TiO$_2$ Thin Films by DC Reactive Magnetron Sputtering*

TiO$_2$ films were grown by DC reactive magnetron sputtering using a titanium disc (Goodfellow, Huntingdon, England, 99.99% purity) with 64.5 mm of diameter and 4 mm of thickness as sputtering target. In order to achieve a base pressure of $10^{-4}$–$10^{-5}$ Pa (before introducing the sputtering gas) a turbomolecular pump (Pfeiffer TMH 1001, Pfeiffer Vacuum GmbH, Asslar, Germany) was utilized. Before the sputter-deposition step of the films, a movable shutter was placed between the target and the supports. The target was pre-sputtered in Ar atmosphere during 2 min to remove the target surface's oxidation. The

target-to-support distance was fixed at 100 mm. The gases used in the deposition were Ar and $O_2$ (both 99.99%, Air Liquide, Paris France) and separate needle valves controlled their pressures. $TiO_2$ deposition was performed in 100% $O_2$ atmosphere. For the $TiO_2$ film growth, the working pressure was 2 Pa, the sputtering power was 520 W, and the deposition time was 25 min (Table 2).

Prior to the depositions, the supports were washed successively in isopropanol and deionized water for 5 min at each step and dried with nitrogen stream to remove any organic contamination.

### 2.4. Morphological Characterization

The morphology observation was achieved by using a field emission scanning electron microscope (FEG-SEM JEOL 7001F) operating at 15 keV. In order to prevent charge buildup, a thin chromium film was coated on the film's surface prior to SEM analysis. The film's thickness was evaluated from the SEM cross-sectional images. Thus, the measured thickness is given by Equation (1) since the sample position presents an angle in relation to the axis of incidence of the electron beam, which can be inferred through the trigonometric geometry involved. The aforementioned correction was estimated through:

$$d_{SEM} = \frac{d_{obs}}{cos\alpha} \tag{1}$$

where, $d_{SEM}$ is the real thickness, $d_{obs}$ is an average of the measured thickness values estimated from the cross-section images and $\alpha$ is the beam incident angle (20°).

The electrical analysis of aqueous matrices was performed with a Solartron 1260 Impedance Analyzer (Solartron Analytical, AMETEK scientific instruments, Berwyn, PA, USA) in the frequency range of 1 to $10^6$ Hz, applying an AC bias voltage of 25 mV, through the measurement of the impedance spectra of these sensor devices, when immersed in the aqueous matrices with different TCS concentrations. To avoid contamination of the sensor devices, the impedance spectra of thin films' sensors were registered for the TCS experimental matrices in a sequential order for increasing concentrations from 0 to $10^{-5}$ M. All measurements were prepared at room temperature (T = 25 °C).

### 2.5. Data Treatment

The Principal Component Analysis (PCA) was performed, considering the normalized (Z-Score normalization $\frac{value-\mu}{\vartheta}$, being $\mu$ and $\vartheta$, the mean value and the standard deviation of the samples, respectively) impedance spectroscopy data, to reduce the size of data and to achieve a new space of orthogonal components, in which different concentration patterns can be perceived and clarified. In addition, an array of sensors, constituted by all produced thin films, was analyzed as an e-tongue for TCS detection in the target matrices.

Linear Discriminant Analysis (LDA) Cross-Validation was performed by Excel XL-STAT Programme (Addinsoft, Paris, France). To cross-validate the PCA data, in the LDA, all variables (different TCS concentrations per type of sensor) for both milk-based emulsions were used in the complete mode and the tolerance was set as 0.001. The response based on impedance measurements data matrix was transformed into canonical patterns. The Mahalanobis distances of each individual TCS concentration to the centroid in a multidimensional space were calculated [32] (Tables S1–S8 in Supplementary Materials).

## 3. Results and Discussion

### 3.1. Sputtered Thin Films' Characterization

The thin films' characterization, using FEG-SEM, allowed the total understanding of the "sensorial" impedance response. The sensorial response is related to the morphology, structure and properties of the thin films and their subsequent interactions with the target molecules' physical-chemical behavior on the aqueous matrices [33]. Figure 1 displays the SEM images and the corresponding cross-section images of the thin films under study.

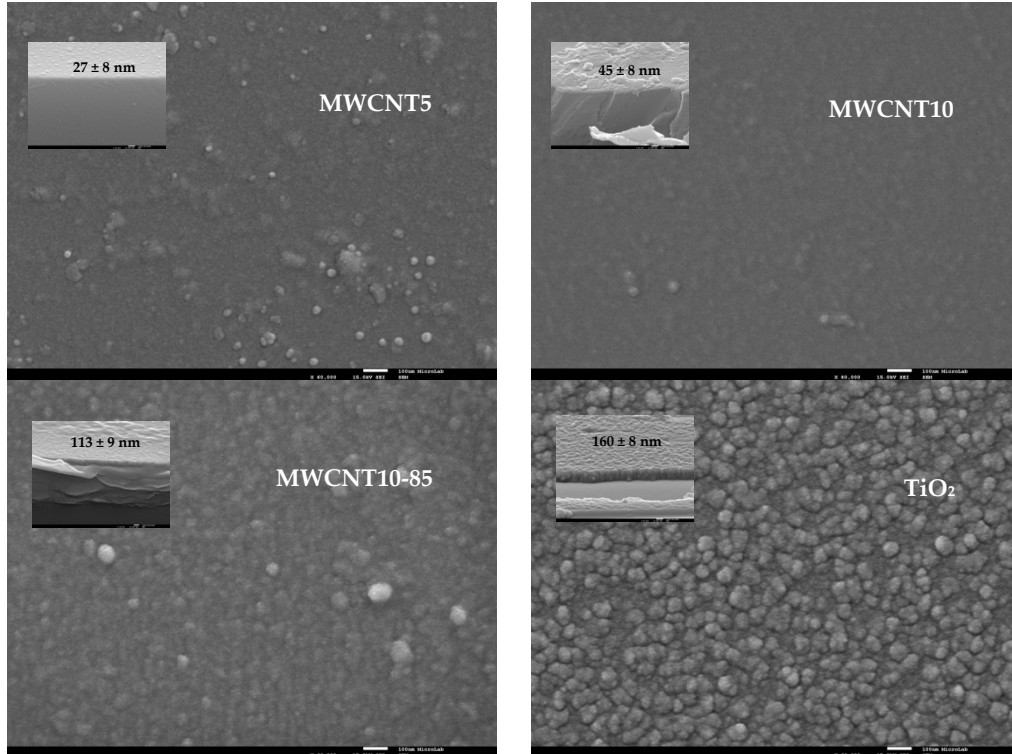

**Figure 1.** Scanning electron microscope (SEM) image of MWCNT5, MWCNT10, MWCNT10-85 and TiO$_2$: surface and cross-section (as inset).

According to Figure 1, the gold IDE supports are covered by highly agglomerated films for both sputtering deposition techniques. In the case of MWCNTs-based thin films, the grains increase with the deposition time and with the applied sputtering power, resulting in a lateral average size around, e.g., 50 and 100 nm for MWCNT10-85. On the other hand, TiO$_2$ films present agglomerates of grains or particulates dispersed throughout the support surface with a 'blooming flower-like' appearance (average lateral size around 80–100 nm), a characteristic of this type of films. This observation is in good agreement with the ones reported recently elsewhere, which were produced in similar conditions [18]. The film's thickness was obtained from FEG-SEM cross-section depicted in the inset of Figure 1. As can be detected, the thickness increases from 27 up to 160 nm, in the mentioned order for the MWCNT5, MWCNT10, MWCNT10-85 and TiO$_2$.

*3.2. Sensor Response: Impedance Spectroscopy Measurements*

As established by Taylor and Macdonald [34], the electrical properties of each thin film, grown on the IDE, when immersed in the aqueous sample, depend on the thin film characteristics used as a sensitive layer and of the double layer formed on the surface of all thin films. Thus, it can be detected in Figure S1 in the Supplementary Data, the impedance spectra of the four magnetron sputtered thin films under study, when immersed in different aqueous matrices doped with increased TCS concentration. Distinctive footprints were observed according to the experimental aqueous matrices and sensor devices. Analogous behavior was noticed for MWCNT10-85 for all matrices. Both MWCNT10 and TiO$_2$ sensors show a comparable behavior, but different electrical curves are exhibited when immersed into tap water or mineral water. The MWCNT5 sensor demonstrates, for all matrices, distinctive footprints.

Figure S1 (Supplementary Materials) further validates that the impedance could be used as a transducing variable, when the aim is to acquire spectra in TCS solutions prepared into the experimental matrices. Following that, to accomplish a more detailed analysis and

to attain the trends of each sensor and understand the pronounced sensitivity to discriminate different TCS' concentrations, a normalized response was performed (Equation (2)):

$$\frac{PP(C) - PP(0M)}{PP(0M)} \tag{2}$$

where *PP(C)* denotes the physical property obtained at given TCS concentrations, and *PP(0 M)* corresponds to the physical property measured at a reference solution for the different type of matrices (TCS 0 M, in tap water (T), mineral water (MW), milk-based prepared with tap water (MT) or mineral water (MMW)). Accordingly, using impedance as one of the sensor's physical properties (PP) at a constant frequency, an increasing or decreasing monotone as a function of TCS concentration can be spotted. Moreover, to allow the operational TCS measurements, a tendency in some features as function of concentration must be achieved, for example, at a given frequency. Figure 2 presents examples of the normalized PP spectra to fixed frequencies, for the different thin films and type of aqueous matrices. These fixed frequency values were selected regarding the frequency where the TCS effects on each type of sensor are better represented.

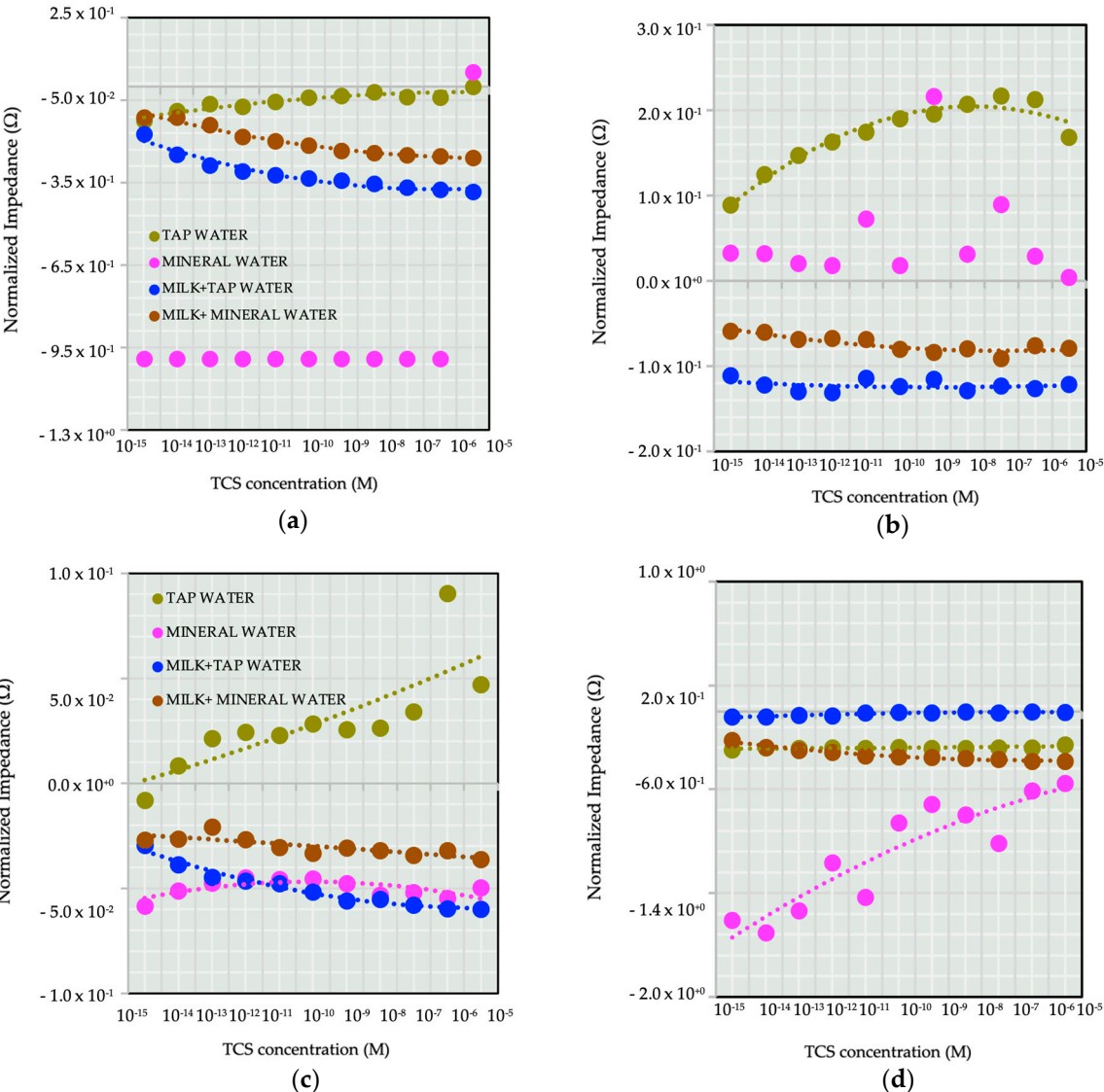

**Figure 2.** Normalized impedance (Ω) versus TCS concentrations at fixed frequencies: (**a**) MWCNT5 at 10 Hz; (**b**) MWCNT10 at 100 Hz; (**c**) MWCNT10-85 at 10 Hz; (**d**) TiO$_2$ at 1000 Hz when immersed in tap water, mineral water, milk-based solution prepared with tap water or mineral water. The lines in the graphs are only guidelines.

It should be noted that it was not possible to pinpoint exactly the most sensitive sensors which presented a clear trend in all matrices under study. However, from Figure 2, it is evident to see the influence on the performance of each sensor in the different matrices. For tap water, MWCNT5 and MWCNT10 (see Figure 2a,b) sensors showed analogous behavior, being considered sensitive to detect the target molecule into this matrix. Both thin films were sputtered with similar conditions, and therefore, the morphology is expected to be comparable (Figure 1). As far as TCS detection in MW is concerned, the electrical response was challenging to attain, due to the low conductivity of the medium, together with the thin film characteristics, which complies with the results found by Magro et al., (2020) [17]. On this matrix, a saturation of the thin film layer up to $10^{-11}$ M TCS concentration seems to occur. Thus, the impedance measurements at fixed frequencies either showed no trend or rise with the TCS concentration, depending on the type of sensor. For MT, MWCNT10 and TiO$_2$ (Figure 2b,d), sensors appeared to be unresponsive (no detectable tendency) to the TCS presence in the studied concentrations. Furthermore, analyzing the data plotted individually and only concerning data discrimination, the most "efficient" sensor seemed to be MWCNT5 at 10 Hz and TiO$_2$ at 1000 Hz (see Figure 2a,d) for the final target milk-based solutions, MT and MMW, respectively. It should also be referred that these matrices have higher quantities of anions when compared to the other two matrices (Table 1, where MT and MMW conductivity takes a value of up to 1.5 mS cm$^{-1}$), which directly contributes to the electrical charge enhancement that can flow between electrodes. Finally, from Figure 2, the impedance behavior of the different types of films can be analyzed and it is possible to observe that MWCNTs thin films presented the following trend: in water solutions (T and MW), the impedance values increase within increasing concentrations, and in milk-based solutions (MT and MMW), the impedance values decrease within increasing concentrations.

Moreover, the sputtered thin films showed to be more sensitive in the discrimination of the TCS in the milk-based matrices, when compared with water matrices. Firstly, TCS is in its non-ionized form in all the matrices, since TCS pKa is higher than solution pH values, thus, there would be less interaction between TCS and charged particles present in the solution, facilitating their interaction with the support and leading to a higher adsorption rate. However, milk is a highly complex medium and includes carbohydrates such as sugar (lactose), fat, citrate, nitrogen in the form of a protein (casein) and non-protein nitrogenous compounds, and mineral salts [19]. Thus, taking into account the high lipophilicity of the TCS, its dissolution in the milk-based matrices would be favored. This characteristic may promote slow movements of TCS molecules within the solutions as they might be bound to the fat particles. Consequently, it would be more difficult for TCS to move away from the support interface, leading to greater interaction with it and resulting in a higher adsorption rate. In fact, Yu et al., (2015) stated that the most influential adsorption mechanisms in CNT thin films are cavity formation, hydrogen bond acidity interactions and π–π electron interactions [35]. Additionally, in Gil et al., (2018), the authors showed that the adsorption between CNT and compounds is predominantly dependent on Van der Waals forces and electrostatic interactions [36]. Additionally, Zhang et al., (2019) [37], further showed that the sorption mechanisms among CNT dispersion and compounds depend on the dispersion properties that the aggregated CNT do not exhibit and also of the principal interactions between MWCNTs and compounds. In the present work, TiO$_2$ thin films revealed a better performance in MMW matrix and this result may be explained by the fact that they are oxides, which may improve its adsorption capacity to TCS compound [27]. On the other hand, for water matrices, the results were less evident, and these results can be explained because CNTs tend to form aggregates in water due to the strong Van der Waals interactions, which can limit their sorption sites' availability for the target compounds [38].

### 3.3. Sensor Capabilities, Sensitivity and Resolution: Electronic Tongue Concept

Principal component analysis (PCA) is one of the most used methodologies in the literature for e-tongues; it is an unsupervised method for determining the relationship between the input data through the comparison of the relative location of the associated so-called principal components in a PCA plot [39]. The PCA was employed to investigate the sensors' electrical attributes to generate different patterns for the range of TCS concentrations. To look for sensor "discrimination" from different TCS concentrations, on distinct matrices, the PCA was carried out after data normalization (Z-Score normalization $\frac{value-\mu}{\vartheta}$, being $\mu$ and $\vartheta$, the mean value and the standard deviation of the samples, respectively). It is important to point out that the characterization related with the best sensor is the one where the deviations in the plot within TCS concentrations are more significant [40]. Thus, Figure 3 displays the PCA concerning impedance normalized data (average of three loops impedance measurements), applied to the aqueous matrices under study. An array of sensors, constituted by MWCNT5, MWCNT10, MWCNT-85 and TiO$_2$ sensors was set in order to understand if the e-tongue concept was capable of "tasting" T, MW, MT and MMW matrices doped with TCS, through the impedance data (Figure 3).

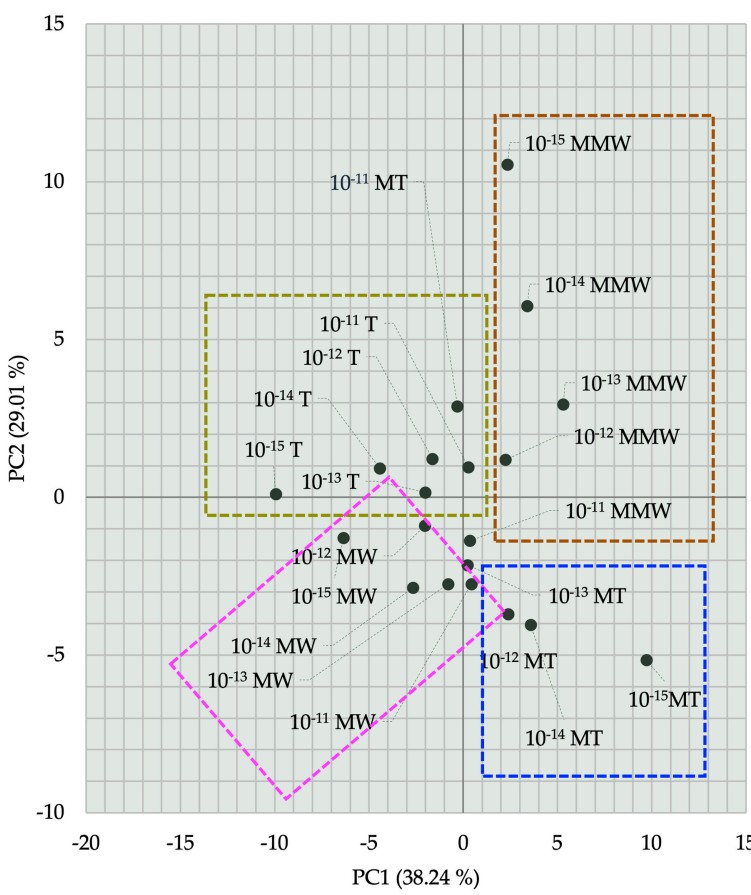

**Figure 3.** Principal components analysis (PCA) plot of TCS concentrations (from $10^{-15}$ to $10^{-5}$ M) distinguished with an electronic tongue sensor, composed by MWCNT5, MWCNT10, MWCNT10-85 and TiO$_2$ sensors, immersed in T, MW, MT and MMW.

The combination of sensor arrays improved the capability of TCS concentration discrimination. The first component PC1, elucidated the greatest data variation and was considered the most important, accounting for more than 38.24% of the variance, when together with PC2, the PCA complies 67.3% of data distribution. In milk-based emulsions, excluding $10^{-11}$ M TCS concentration, the e-tongue clearly distinguished water from milk-based matrices through PC1. The e-tongue places into the positive part of PC1 the matrices with infant formula powder, and in the negative part, the water matrices.

Inside of the distinguished matrices areas, a concentration pattern and tendency, related to TCS concentrations (between $10^{-11}$ and $10^{-15}$ M) is visible only for the data obtained for MMW solution, where eigenvector points tended to decrease throughout the studied concentrations' range.

Moreover, it is important to understand individually the PCA e-tongue response, for the final purpose of triclosan detection in milk-based emulsions. Thus, a singular PCA was plotted for MT and MMW, which are depicted in Figure 4.

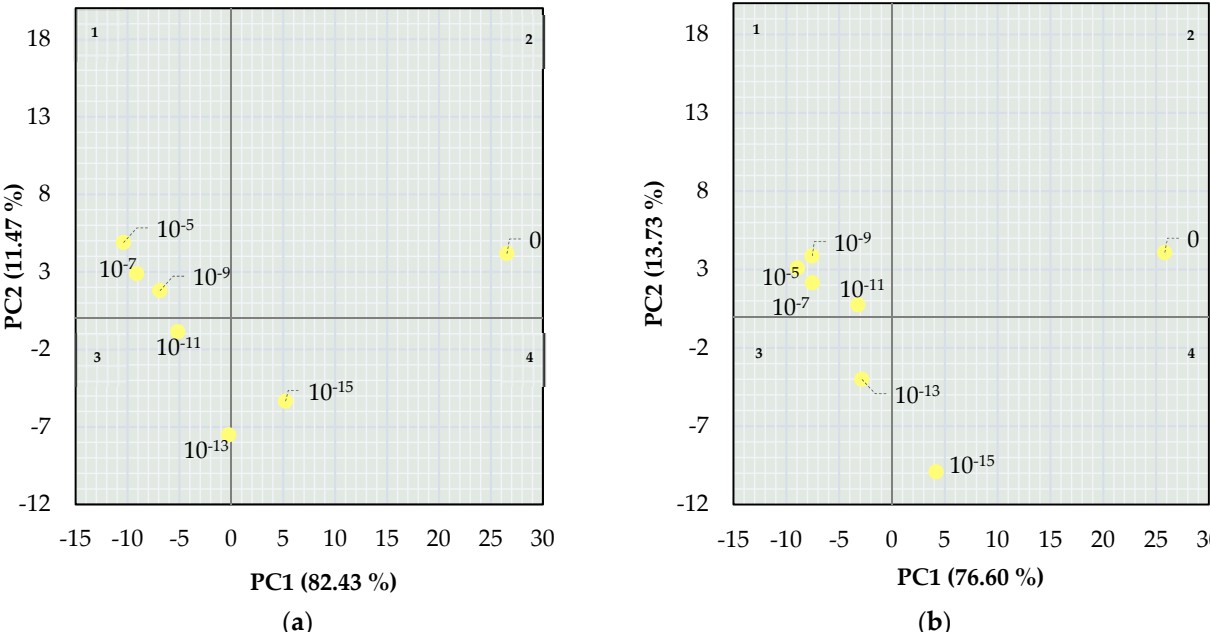

**Figure 4.** Principal components analysis (PCA) plot of TCS concentrations ($10^{-15}$–$10^{-5}$ M) distinguished with an electronic tongue sensor for MT (**a**) and MMW (**b**).

In MT matrix, the first two principal components PC1 and PC2 represented 90.33% of the total variance. This device was capable of detection and semi-quantification since it was able to define an evident trend and pattern within the PCA plot till $10^{-9}$ M of TCS. Additionally, there was a noticeable pattern, where the concentration of the non-doped solutions (TCS = 0 M) might be considered as an outlier point, corresponding to the nearest concentration, in both cases $10^{-15}$ M.

A concentration pattern and trend related to TCS concentrations revealed to be even more pronounced as it can be seen from Figure 4b analysis, where the first two principal components PC1 and PC2 are seen to account for 93.90% of the total variance. As it can be noted in Figure 4a, the data obeyed the same direction, from region 2 (TCS) = 0 M to region 4 where (TCS) > 0 M, demonstrating that the electronics based in the sensor devices were able to discriminate between non-doped and doped MMW, producing an observable pattern and tendency concerning the different concentration values.

To further complement the PCA findings regarding the discrimination capabilities of the developed e-tongue for TCS detection in milk-based solutions, a Linear Discriminant Analysis (LDA) [32,41] was used as a cross-validation method. Accordingly, for each observation, factor scores (shown as the probability to belong to each concentration group) and squared Mahalanobis distances to the centroid of the group were achieved throughout a cross-validation matrix (per sensor) [32]. Therefore, for MWCNT5, MWCNT10, MWCNT10-85 and TiO$_2$ sensors in MT and MMW matrices, the LDA cross-validation revealed that: (1) higher concentrations measured (up to $10^{-12}$ M TCS) may saturate the thin film; (2) occasionally, lower TCS concentrations may not be distinguished as different concentrations ($10^{-15}$ M vs. $10^{-14}$ M TCS); (3) the adequate classification is dependent

on the sensor, as already above-mentioned in the PCA. The confusion matrix showed a cross-classification accuracy of 60.4% (Table S8 in Supplementary Materials).

Furthermore, an analytical performance study was performed to find the best fitting between both milk-based solutions prepared with tap and mineral water. Figure 5 presents the first principal component factor scores as a function of TCS concentrations working as an e-tongue device and a linear regression for MT and MMW data.

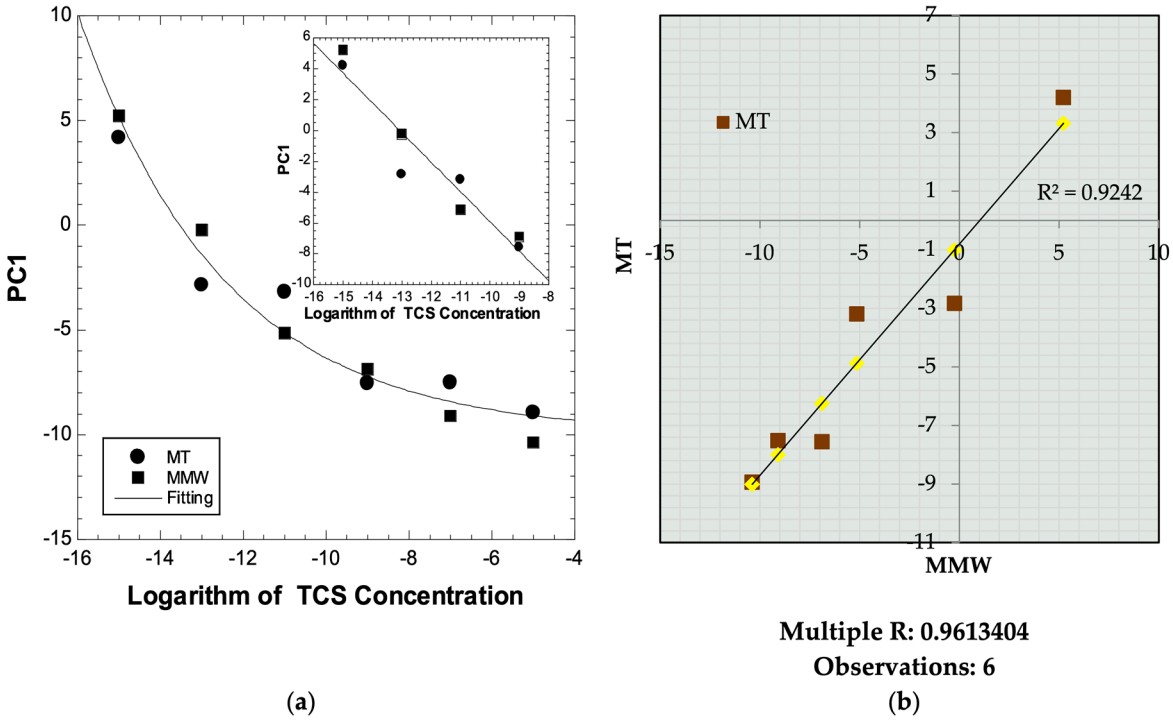

**Figure 5.** (**a**) Principal components PC1 factor scores versus TCS concentrations for MT and MMW e-tongue (data from Figure 4); (**b**) PC1 distributions of MT and MMW e-tongue. Prediction data (adjusted with the MT data) are in yellow. A linear fit to the data is shown as a black line.

Figure 5a shows the exponential decrease of PC1 factor scores with the increase of the logarithm of TCS concentration (from $10^{-15}$ to $10^{-5}$ M) in MT and MMW. This exponential tendency suggests excluding the values of PC1 data that tend to a constant value, i.e., considering only the PC1 values within TCS concentration range of $10^{-15}$ and $10^{-9}$ M, a linear fitting can be attained (see inset of Figure 5a). Considering the data between that concentration range and the PC1 feature plot versus the logarithm of the concentration, the sensor sensitivity can be achieved by the slope of the output characteristic curve (a straight-line fitting) $\frac{\Delta PC1}{\Delta logC} = -1.92 \pm 0.23$. The concentration response of the e-tongue is indicated by this negative value ($-1.9$) of sensitivity [42], demonstrating, from Figure 5a, that the sensitivity decreases with the increase of the concentration. The sensor resolution [43], corresponding to the lowest concentration that can be detected (the limit of detection) and may be found near the lowest used concentration ($C_s$). The $C_s$ value was $10^{-15}$ M, while the minimum value that could be measured was 0.23. Thus, considering $logC = \frac{0.23}{-1.92}$ and $\Delta logC = logC - logCs = 0.1197$, the lower concentration measured is $7.6 \times 10^{-16}$ M, which corresponds to the sensor resolution. Furthermore, it is important to establish the potential of the e-tongue coating materials and its relationship in each type of infant milk formula preparations. To understand this relationship, a linear regression was also carried out for the data depicted in Figure 4. Figure 5b presents, therefore, a linear fit plot for 95% of the confidence level between the PC1 from MT and the PC1 from MMW data. Analyzing the linear regression, MT and MMW distributions are highly correlated, with a correlation coefficient of 0.96 for TCS detection. It should be noted that these

correlations were primarily limited by the few observations, nevertheless, a clear trend may be observed.

## 4. Conclusions

An electronic tongue, comprising of an array of sensors based on sputtered thin films revealed to be able to detect and quantify triclosan trace concentrations, within the range of $10^{-15}$–$10^{-5}$ M, in milk-based matrices, through the impedance spectra measurement of each sensor's device.

The experimental data suggested that from a structural and manufacturing point of view, the choice of the type of thin film or sensorial layer is a critical step for the accomplishment of reliable qualitative and quantitative evaluation. The surface morphology of the thin films depends on the deposition parameters and consequently are key factors on the overall device impedance response. Therefore, for the T, MW, MT, MMW matrices, the sensors which revealed to be the most sensitive in the studied TCS concentration range were MWCNT10 at 100 Hz, MWCNT10-85 at 10 Hz, MWCNT5 at 10 Hz and TiO$_2$ at 1000 Hz, respectively.

From the analysis of the e-tongue data through a PCA, LDA Cross-Validation and linear regression combination, the sensor's ability and potential to distinguish between water and milk-based matrices and to discriminate TCS concentrations using the first principal component was demonstrated. PC1 results for both type of infant milk formula powder preparation samples followed the same behavior with respect to TCS concentration. Values of $-1.9 \pm 0.3$ and of $7.6 \times 10^{-16}$ M for the e-tongue sensitivity and resolution, respectively, were obtained in range of detection from $10^{-15}$ to $10^{-9}$ M. Finally, this study suggests a new perspective for the detection of triclosan traces in complex media by using sputtered thin films of MWCNT5, MWCNT10, MWCNT10-85 and TiO$_2$ as sensing units allowing the continuous monitoring of this compound.

**Supplementary Materials:** The following are available online at https://www.mdpi.com/2079-6412/11/3/336/s1, Figure S1: Impedance spectra of the sensor device with thin films of (a) MWCNT5; (b) MWCNT10; (c) MWCNT10-85; (d) TiO$_2$ immersed in tap water, mineral water, milk prepared with tap water or mineral water at different TCS concentrations; Linear Discriminant Analysis (LDA) obtained with data from the array of sensors (e-tongue) build-up to detect different TCS concentrations in milk-based emulsion (Tables S1–S8 in Supplementary Materials).

**Author Contributions:** Conceptualization, M.S., M.R. and S.S.; methodology, M.S.; software, M.S.; validation, C.M., P.A.R., M.R. and S.S.; formal analysis, C.M., P.A.R., M.R. and S.S.; investigation, M.S. and C.M.; resources, P.A.R., M.R. and S.S.; data curation, M.S.; writing—original draft preparation, C.M., M.R. and S.S.; writing—review and editing, C.M., P.A.R., M.R. and S.S.; supervision, C.M., P.A.R., M.R. and S.S.; project administration, M.R.; funding acquisition, P.A.R., M.R. and S.S. All authors have read and agreed to the published version of the manuscript.

**Funding:** This work was supported by the Project "Development of Nanostructures for Detection of Triclosan Traces on Aquatic Environments" (PTDC/FIS-NAN/0909/2014). The Center for Environmental and Sustainability Research (CENSE) and Centre of Physics and Technological Research (CEFITEC), which is financed by national funds from FCT/MEC (UID/AMB/04085/2019 and UID/FIS/00068/2019). The authors also acknowledge and appreciate the support given to Center for Environmental and Sustainability Research (CENSE) and Centre of Physics and Technological Research (CEFITEC) by the Portuguese Foundation for Science and Technology (FCT) through the strategic project UIDB/04085/2020 and UID/FIS/00068/2020.

**Institutional Review Board Statement:** Not applicable.

**Informed Consent Statement:** Not applicable.

**Acknowledgments:** The authors acknowledge Alexandra B. Ribeiro and Eduardo P. Mateus for their important and critical insights and comments in Margarida Sardinha Master Thesis' defense. The authors would also like to acknowledge Isabel Dias Nogueira for the SEM/EDS analysis.

**Conflicts of Interest:** The authors declare no conflict of interest.

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
