# Peer review of "Magnetron Sputtering Thin Films as Tool to Detect Triclosan in Infant Formula Powder: Electronic Tongue Approach"

_coatings, doi:10.3390/coatings11030336_

Round 1

Reviewer 1 Report

The authors developed impedance-based electronic tongues, and used the electronic tongues to detect triclosan in solutions (tap water, and milk powder solutions). The novelty of the work is the combination of impedance spectroscopy and electronic tongues, it makes the electronic tongues suitable for situations where electrochemical detection becomes difficult, for example when the concentration of the analyte is too low. The article is quite detailed, I don’t have much too comment, just some minor issues, as listed below:

  1. There’re some grammar issues, I’ll list a few locations I noticed and marked below.

The sentence from line 209 to 211 looks strange. Should it be ‘where’ instead of ‘were’ in line 210?  

And at the end of line 213, it might be written as this: However, from Figure 2, it is evident

And the sentence from line 232 to 233, ‘ from Figure 2, it can be analysed the impedance behavior of the different types of films’,  it’s better to be written as ‘from Figure 2, the impedance behavior of the different types of films can be analysed’

  1. The authors wrote that the sensitivity of the sensor is -1.9±0.2 M, and the and resolution is 7.6 x 10-16 M. As a electrochemist, I am not sure what those properties really stand for. We typically use terms like ‘limit of detection (LOD)’, which seems to be similar to the resolution. Then I do not know what the sensitivity stands for, and why it is a negative value. Could the authors clarify this part a bit more, and maybe add some citations? It would help readers like me to better understand the concepts here.

Reviewer 2 Report

I read this study with great interest. 

I have some comments:

  • PCA is an explorative analysis. For this kind of data, at least for internal validation, I suggest to perform linear/canonical discriminant analysis with Cross Validated Accuracy value calculation among groups. It would strenghten your observations.
  • Did authors perform sample size estimation? Please specify it.
  • Minor english errors throughout the paper. 

Round 2

Reviewer 2 Report

Authors replied to all my queries. 

I believe that now this article can be accepted.